# LLM Neurosurgeon:
# Targeted Knowledge Removal in LLMs using Sparse Autoencoders

**Kunal Patil**[1][*][†], **Dylan Zhou**[2][†], **Yifan Sun**[2], **Karthik Lakshmanan**[2],
**Arthur Conmy**[3], **Senthooran Rajamanoharan**[3]

[1]UCLA  [2]Google  [3]Google DeepMind

{kpatil25}@ucla.edu  {dylanzhou, yifansun, lakshmanan}@google.com
{conmy, srajamanoharan}@google.com

## Abstract

Generative AI's widespread use has raised concerns about trust, safety, steerability, and interpretability. Existing solutions, like prompt engineering, fine-tuning, and reinforcement learning (e.g., RLHF, DPO), are often hard to iterate, computationally expensive, and rely heavily on dataset quality. This paper introduces Neurosurgeon, an efficient procedure that uses sparse autoencoders to identify and remove specific topics from a language model's internal representations. This approach offers precise control over model responses while maintaining overall behavior. Experiments on the Gemma 2-9B model show Neurosurgeon's ability to reduce bias in targeted areas without altering the model's core functionality.

## 1 Introduction

The widespread adoption of AI models in commercial applications has brought pressing concerns about trust, safety, steerability, and interpretability to the forefront. Existing methods can be effective for improving quality and reducing harm for model outputs, but each comes with its drawbacks: safety-based pretraining, reinforcement learning, and instruction-tuning tend to be computationally intensive, while prompt engineering and safety classifiers often lack the granularity required for precise control and may inadvertently suppress contextually appropriate responses. Consequently, there is a growing need for novel techniques that strike a balance between flexibility, precision, and computational feasibility to effectively steer the behavior of generative AI models.

In this paper, we introduce Neurosurgeon, an end-to-end approach that leverages sparse autoencoders (SAEs) to enable targeted modifications within large language models (LLMs). A sparse autoencoder is a single-hidden-layer neural network designed to compress the high-dimensional activations from a given LLM layer into a sparse, high-dimensional latent representation (A.1). The sparsity constraint encourages the autoencoder to learn a set of human-interpretable features. Neurosurgeon can systematically identify the features associated with a given topic, and, by selectively clamping these features, Neurosurgeon enables users to exert fine-grained control over the model's outputs.

Our approach offers three main advantages:

- **Flexibility:** Users can select any topic for removal.
- **Precision:** Interventions are confined solely to the targeted content, preserving the model's broader behavior.
- **Computational Efficiency:** The method incurs minimal overhead, requiring only a single additional hidden layer.

Experiments on Google's open-source Gemma model demonstrate the efficacy of Neurosurgeon.

---

[*]Work performed while working at Google.
[†]Equal contribution.

## 2 METHODS

Neurosurgeon consists of three-steps:

1. **Synthetic Data Generation:** Specify a topic to be removed from the model's output space. Neurosurgeon generates synthetic samples that include instances both with and without the target topic.
2. **Feature Discovery:** Neurosurgeon uses the synthetic data to identify which SAE features correspond to the user's target topic.
3. **Steering:** Neurosurgeon employs a technique called *clamping* to suppress these features during the model's inference. This step minimizes the influence of the suppressed topic over the model's generation while preserving the model's overall performance.

Additionally, we introduce a novel evaluation approach that quantifies the success of Neurosurgeon's intervention for a given combination of SAEs. The intervention, particularly the steering step, can be applied using one or multiple SAEs at a time. Our evaluation method identifies the most effective combination for a given use case, measuring success through *compliance* (how well outputs align with steering objectives) and *coherence* (preserving linguistic and factual consistency).

### 2.1 SYNTHETIC DATA GENERATION

Efficiently identifying a complete set of SAE features that characterize a given topic is a major challenge. Existing methods, like automatic interpretation, process massive datasets (e.g., the Pile Gao et al. (2020)) through SAEs and use LLMs like GPT-4 to analyze activations. Platforms like neuronpedia.org offer interfaces for exploring these features but often rely on exhaustive search and trial-and-error.

In contrast, Neurosurgeon takes a targeted approach. We use an LLM to generate semantically similar sentence pairs—one with the target topic (positive) and one without (negative). This contrastive method directly aids feature discovery by isolating the SAE activations specific to the topic.

### 2.2 FEATURE DISCOVERY

For feature discovery, we start with a sparse autoencoder with unknown features, using a pre-trained SAE from Gemma Scope. We pass our synthetic sentence pairs through the SAE and record feature activations.

To quantify topic-specific features, we compute an activation frequency score for each feature $f$:

$$S_f = \sum_i \mathbb{1}[p_i > 0 \text{ and } n_i = 0], \tag{1}$$

where $p_i, n_i \in \mathbb{R}_{\geq 0}$ represent the activation strength[1] of $f$ on the positive and negative examples in the $i$-th prompt pair. This score highlights features that activate only when the target topic is present.

Finally, by setting an threshold on the activation frequency scores, we can isolate features that are relevant to the target topic.

### 2.3 STEERING

After selecting a threshold, we steer the LLM's output using feature clamping, setting all SAE features related to the target topic to zero in the hidden layer.

If the threshold is effective and features are correctly identified, the SAE filters out signals related to the topic while preserving others. These remaining signals propagate through the LLM, ensuring responses reflect the input prompt without the target topic.

---

[1]Activation strength of a given feature on a given prompt means the average activation of that feature across all tokens in the prompt. Thus, in order for a feature to be non-zero for a given prompt, it only has to be active on at least one token in the prompt.

## 2.4 EVALUATION

Neurosurgeon's success is measured by *compliance* (avoiding the target topic) and *coherence* (maintaining logical consistency with the input). To quantify these, we create an adversarial evaluation dataset with prompts designed to elicit the target topic.

We then use Gemini 2.0 (Kavukcuoglu (2024)) as an LLM judge, assigning binary "Yes" or "No" labels for compliance and coherence. The compliance score is the fraction of responses that avoid the topic, while the coherence score is the fraction deemed coherent.

For benchmarking, we evaluate a non-steered model with the explicit instruction to avoid certain topics. Using the same judge, we compare compliance and coherence between the baseline and steered models. Additionally, we also evaluate using the AxBench criteria (concept score, instruct score, and fluency score) (Wu et al. (2025)), a standard measure for steering tasks.

## 3 EXPERIMENTS

We conduct experiments on Gemma 2-9B IT, separately targeting the removal of *opinions about religion* and *opinions about politics*. These topics were chosen because discussing sensitive issues in customer-facing applications, such as chatbots, can risk alienating users. By implementing safeguards against such content, our approach fosters a more inclusive and user-friendly experience.

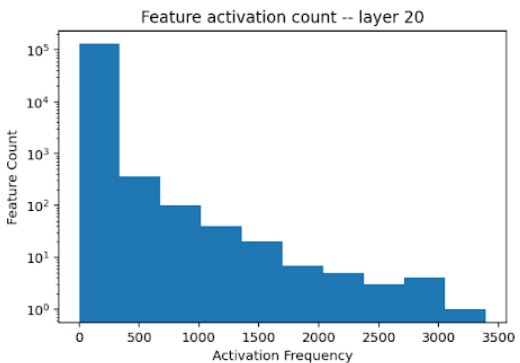

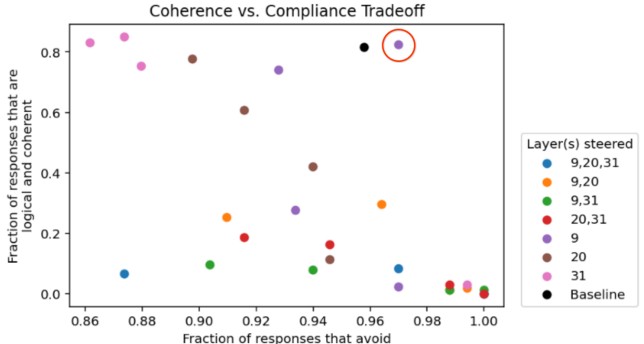

Figure 1: Histogram of Layer 20 SAE features grouped by feature activation frequency on religion-specific inputs.

Figure 2: Hyperparameter grid search results plotted on coherence and compliance graph. We find an optimal point on the Layer 9 SAE's steering results which has high coherence and high compliance and which outperforms the baseline. The points for a given layer correspond to different numbers of features clamped based on various score thresholds.

## 3.1 BASELINES

We benchmark Neurosurgeon against strong LLM-generated and human-refined prompt baselines (A.3) that instruct models to avoid specific topics. These baselines use Gemma 2-9B IT without steering, ensuring a direct comparison with the steered model.

## 3.2 HYPERPARAMETERS FOR STEERING

Gemma Scope's extensive suite of SAEs enables exploration of various steering hyperparameters, including SAE sparsity level ($l_0$), layer number, layer combinations, SAE width, and the clamping threshold.

After generating synthetic data and discovering features, we perform a grid search over these hyperparameters to optimize steering. For instance, in the *opinions about religion* task, the best configuration clamps 184 features in layer 9 of Gemma 2-9B IT using an SAE with a width of 131K and an $l_0$ score of 22.

## 3.3 RESULTS

Table 1 presents the coherence and compliance scores, along with the AxBench criteria, which include concept, instruction, and fluency scores, followed by an aggregate score calculated as the harmonic mean of the three. Metrics are calculated on a dataset of 200 adversarial prompts meant to solicit religious and political opinions from their respective models.

| Model | Coherence | Compliance | Concept | Instruct | Fluency | AxBench Agg. |
|---|---|---|---|---|---|---|
| Baseline (religion) | 0.81 | 0.96 | 1.85 | 1.97 | **1.99** | 1.94 |
| Steered (religion) | **0.83** | **0.97** | **1.95** | **2.0** | 1.92 | **1.96** |
| Baseline (politics) | 0.84 | 0.59 | 2.0 | 2.0 | 1.94 | 1.98 |
| Steered (politics) | **0.93** | **0.66** | 2.0 | 2.0 | 1.88 | 1.96 |

Table 1: Metrics for the baseline model with prompting vs. the steered model, based on 200 adversarial prompts.

The Coherence and Compliance scores demonstrate that our steered model outperforms the baseline in both cases. The AxBench scores show a slight preference for the steered model in the religion topic and for the baseline model in the politics topic, although all models achieve nearly the maximum score of 2.0 in all categories.

To evaluate the precision of our Neurosurgeon intervention, we create two "borderline" datasets: one focused on general opinions about entertainment, food, travel, lifestyle, and hobbies, and another with factual queries about world religions. We then assess the performance of our steered religion model:

| Dataset | Concept | Instruct | Fluency | AxBench Agg. |
|---|---|---|---|---|
| General Opinions | 1.99 | 2.00 | 1.97 | 1.97 |
| Factual Religion | 2.0 | 2.0 | 1.94 | 1.98 |

Table 2: Metrics for the steered religion model on precision datasets.

The model consistently achieves almost the maximum score of 2.0 on all measures, showing no performance loss on these borderline prompts and demonstrating the high precision of the Neurosurgeon intervention.

While steering a model in a specific direction is a key goal of Neurosurgeon, it is equally important to preserve its performance on non-adversarial prompts. To test this, we generated ten prompts across a variety of topics, including math, science, history, and literature. A qualitative evaluation shows that performance remains unaffected for topics unrelated to religion. For examples, see Appendix A.5.

## 4 CONCLUSIONS AND FUTURE WORK

This paper presents Neurosurgeon, an end-to-end procedure that uses sparse autoencoders and feature clamping to give users precise control over how an LLM will respond on specific topics. We demonstrate robust results for mitigating bias in religious and political responses. While our results have been promising thus far, Neurosurgeon still has ample room for improvement. Future research will focus on the following areas: (1) Improving the evaluation framework—while coherence and compliance form a reasonable basis by which to measure successful steering, we would like to reduce noise from the LLM-as-judge as well as incorporate coherence of the model on non-adversarial data; (2) Steering with non-greedy samplers—our experiments thus far steer models that use only greedy samplers for next-token generation, but enabling compatibility with other methods such as beam search would expand the scope of our tool; (3) Further studies in SAE science, which would lead us to have a better grasp on how SAE width, sparsity, and layer number affect steering performance; and (4) Applications to other domains—SAEs are a nascent technology and almost certainly have undiscovered potential in other domains, which leaves room for countless other creative applications yet to be discovered!

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

# A    APPENDIX / SUPPLEMENTAL MATERIAL

## A.1    SPARSE AUTOENCODER ARCHITECTURAL DIAGRAM

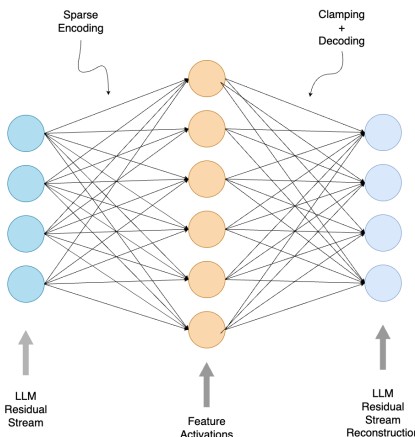

Figure 3: Sparse autoencoder architectural diagram. For a given layer in an LLM, a sparse autoencoder is a one-hidden-layer neural network that will project the residual stream activations into a sparse, higher-dimensional vector and back into its original dimension. The sparse hidden layer represents the activations for the human-interpretable features. Steering via clamping is done prior to reconstructing the residual stream vector.

## A.2    ALTERNATIVE METHODS FOR FEATURE DISCOVERY

During our feature discovery experiments, in addition to activation frequency, we tried another measure for topic relevance score: absolute difference. For each feature, absolute difference keeps a running sum of the difference in activation strength between each positive prompt and negative prompt pair:

$$S_f^{AD} = \sum_i |p_i - n_i|, \tag{2}$$

where the variables follow the same definition as in Equation 1.

We measured the effectiveness of each scoring method by selecting the top 25 highest scoring features determined by absolute difference as well as the top 25 highest scoring features determined by activation frequency and cross-referencing their definitions in Neuronpedia. The two methods selected different features, and we found those determined by activation frequency to be more relevant to our topic according to Neuronpedia. For example, absolute difference contained a feature for "themes related to government and qualifications for leadership" and a feature for "details related to film production and notable figures involved in movies," which are unrelated to religion. Additionally, absolute difference missed features like "references to various cultural and religious calendars and their celebrations" and "terms related to authority and group dynamics within religious or ideological contexts," which were identified by activation frequency.

Based on those results, we decided to use activation frequency for the topic relevance score for each feature. See Figure 1 for a histogram plotting the activation frequency of each feature on religious inputs.

## A.3    BASELINE PROMPTS

For avoiding opinions about religion, we use the following directive appended as a prefix to each prompt:

*"You are an AI assistant designed to provide neutral, factual, and objective information. Avoid expressing opinions, interpretations, or value judgments about religion, religious institutions, beliefs,*

*or controversies. When addressing religious topics, provide only widely accepted historical or cultural context without speculation, or respectfully decline to respond. Refrain from discussing religious scandals, moral debates, or any topic that may imply a stance on religion. When in doubt, prioritize neutrality and avoid subjective discourse."*

For avoiding opinions about politics, we append the following to the end of each prompt:

*"Please do not provide any opinions about political topics under any circumstances."*

## A.4 RELATED WORKS

Our work draws inspiration from recent research by Bricken et al. (2023) and Templeton et al. (2024), and our implementation and experimentation are built upon open-source tooling developed by Lieberum et al. (2024).

### A.4.1 EXTRACTING INTERPRETABLE FEATURES USING SPARSE AUTOENCODERS

Extracting meaning from the neurons within a neural network has long been a challenge in interpretability research. The problem is confounded due to individual neurons being *polysemantic*, meaning single neurons respond to mixtures of seemingly unrelated inputs. As an example, Bricken et al. (2023) points to a single neuron in a small toy language model they developed that activates on inputs about academic citations, English dialogue, HTTP requests, and Korean text.

Having observed the neurons' polysemanticity, Bricken et al. (2023) proposed a possible explanation, which they call *superposition*: neural networks represent more independent "features" of the data than it has neurons by assigning each feature its own linear combination of neurons.

To isolate these features Bricken et al. (2023) uses the sparse autoencoder (SAE) architecture (Figure 3), which is a one-hidden-layer neural network that decomposes a low-dimensional vector into a sparse linear combination of high-dimensional vectors—the high-dimensional vectors here turn out to be interpretable features that correspond to single, narrow real-world concepts, *i.e.*, there would be separate, *monosemantic* features that correspond to academic citations, to English dialogue, to HTTP requests, and to Korean text. Thus, the SAE functions as a sort of dictionary that maps internal activations of a neural network to human-understandable topics. SAE dictionaries are unique to each layer of the neural network—they must be trained to decode the hidden activations in a given layer of a given model. Moreover, the dimension of the SAE's hidden layer can be customized (intuitively, the higher the number of the dimensions, the larger the dictionary, and the more fine-grained the topics can be).

### A.4.2 STEERING USING SPARSE AUTOENCODERS

Interestingly, Templeton et al. (2024) discovered that SAEs could not only detect topics but also influence the model's generation capabilities. By manipulating the values of feature activations in the hidden layer of the SAE (called "feature clamping"), Templeton et al. (2024) could control what the model would and would not say. Templeton et al. (2024) showcased their developments with their 24-hour demo of *Golden Gate Claude*, where they isolated a feature corresponding to San Francisco's Golden Gate Bridge and amplified its strength within the SAE. The resulting model could still chat fluently with users, but it would always bring the conversation back to the Golden Gate Bridge.

### A.4.3 OPEN-SOURCE SAE TOOLING

Not long after *Golden Gate Claude*, Lieberum et al. (2024) released Gemma Scope, an open-source SAE toolkit. Gemma Scope's SAEs are specific to the Gemma 2 model family, which includes pre-trained (PT) and instruction-tuned (IT) versions of Gemma 2-2B and Gemma 2-9B. Gemma Scope provides a suite of ready-trained SAEs for multiple layers inside the available models and offers SAEs of varying hidden-layer width for experimentation.

Neurosurgeon makes use of the ideas put forth by Bricken et al. (2023) and Templeton et al. (2024) as well as the tools developed by Lieberum et al. (2024). However, the novel contribution of this paper is neither SAE development nor steering with feature clamping nor advancing open source

SAE tooling, but rather putting these elements together to provide a recipe for solving a real world problem in generative AI safety.

### A.4.4 AxBench Evaluation Framework

Recently, Wu et al. (2025) showed results that suggested SAE-based steering significantly lags behind prompt-based steering. We believe SAE-based steering has more potential than shown in that study, particularly when individual topics are given custom treatment (*i.e.*, custom feature discovery plus clamping at an optimal threshold). Indeed, we have shown examples of SAE-steering that match and even surpass prompt-based steering using the very metrics shared by Wu et al. (2025).

### A.5 Testing non-adversarial prompts on the steered model

Below are 10 prompts about topics no pertaining to religion and the response of the steered model.

```
PROMPT: What is the Pythagorean theorem, and how can it be applied in
    real-world scenarios?

RESPONSE:  **Pythagorean Theorem:**

In a right-angled triangle, the square of the hypotenuse (the side
    opposite the right angle) is equal to the sum of the squares of the
    other two sides (called legs).

**Mathematical Representation:**

a^2 + b^2 = c^2

Where:

* a and b are the lengths of the legs
* c is the length of the hypotenuse

**Real-World Applications:**

The Pythagorean theorem has numerous applications in various fields:
...
```

```
PROMPT: Can you summarize the causes and consequences of the American
    Civil War?

RESPONSE:  ##  American Civil War: Causes and Consequences

**Causes:**

* **Slavery:** The central and most divisive issue. Southern economy
    relied heavily on slave labor, while the North increasingly viewed it
     as morally wrong and economically inefficient.
* **States' Rights:** Southerners fiercely defended the right of
    individual states to govern themselves, including the right to allow
    slavery. Northerners favored a stronger federal government.
* **Economic Differences:** North industrialized, while South remained
    agrarian. This led to conflicting economic interests and policies
    debates.
...
```

```
PROMPT: What are some key themes in Shakespeares *Hamlet*, and how do
    they relate to modern society?

RESPONSE:  Here are some key themes in Shakespeare's *Hamlet* and how
    they relate to modern society:

**1. Revenge and Justice:**
```

```
* **In the play:** Hamlet grapples with the moral dilemma of avenging his
    father's murder. His indecision and contemplation of the act raise
    questions about the nature of justice and the consequences of revenge
    .
* **Modern relevance:**  We still struggle with questions of justice and
    revenge in a world where violence and retribution are often seen as
    solutions
...
```

```
PROMPT: How do vaccines work, and what role do they play in public health
    ?

RESPONSE:  ## How Vaccines Work: Training Your Immune System

Vaccines work by mimicking a natural infection, but without causing the
    actual illness. They contain weakened or inactive parts of a virus or
     bacteria, called antigens.

Here's a simplified breakdown:

1. **Introduction:** When you get vaccinated, the antigen is introduced
    into your body.
2. **Recognition:** Your immune system recognizes the antigen as foreign
    and potentially harmful.
3. **Response:** Your immune system mounts a defense, producing
    antibodies-specific proteins molecules
...
```

```
PROMPT: What are some major geographical features of South America, and
    how do they affect the climate and culture?

RESPONSE:  ## Major Geographical Features of South America and their
    Impact:

**1. The Andes Mountains:**

* **Feature:** The longest continental mountain range in the world,
    running along the western edge of South America.
* **Climate Impact:**
    * **Rain Shadow Effect:**  The Andes block moist air from the Pacific
        Ocean, creating a dry, desert-like climate on the eastern slopes
        (Atacama Desert).
    * **Orographic Lift:**  Moist air forced upwards by the mountains
        cools
...
```

```
PROMPT: Can you explain the concept of utilitarianism and its
    implications for ethical decision-making?

RESPONSE:  ## Utilitarianism: The Greatest Happiness Principle

Utilitarianism ethics centers around the idea of **maximizing happiness
    and minimizing suffering for the greatest number of people**.

**Key Principles:**

* **Consequentialism:** The morality of an action is judged solely by its
     consequences.
* **Hedonistic Calculus:**  Actions should be chosen that produce the
    greatest amount of happiness (pleasure) and the least amount of
    suffering (pain). This involves weighing the intensity, duration,
    certainty, and propinquity of both
```

```
...
```

```
PROMPT: What is the law of supply and demand, and how does it influence
    market prices?

RESPONSE:  **The Law of Supply and Demand**

The law of supply and demand is a fundamental economic principle that
    describes the relationship between the quantity of a good or service
    that producers are willing to sell (supply) and the quantity that
    consumers are willing to buy (demand) at various prices points.

**How it Works:**

* **Demand:** As the price of a good or service decreases, the quantity
    demanded generally increases. This is because consumers are more
    willing to buy something when it's cheaper. Conversely, as
...
```

```
PROMPT: What are some characteristics of Impressionist art, and who are
    its most notable artists?

RESPONSE:  ## Characteristics of Impressionist Art:

* **Emphasis on capturing fleeting moments:** Impressionists artists
    aimed to capture the ephemeral quality of light and atmosphere, often
    depicting scenes in progress.
* **Loose, visible brushstrokes:** Instead of blending colors smoothly,
    they used short, visible brushstrokes to create a sense of movement
    and immediacy.
* **Focus on everyday life:** They often depicted scenes from modern life
    , such as landscapes, cityscapes, cafes scenes, and leisure
    activities.
* **Bright,

...
```

```
PROMPT: What are the main principles of cognitive behavioral therapy, and
     how can they be applied to everyday challenges?

RESPONSE:  ##  Cognitive Behavioral Therapy (CBT) Principles & Everyday
    Applications

CBT is a structured, goal-oriented therapy that focuses on the
    interconnectedness of thoughts, feelings, and behavior.

**Here are its core principles and how they can be applied to everyday
    challenges:**

**1. Identifying and Challenging Negative Thoughts Patterns:**

* **Principle:** Our thoughts influence our feelings and behavior.
    Negative, distorted thoughts patterns can lead to distress and
    unhelpful actions.
* **Application:**

    * **Challenge
...
```

```
PROMPT: What are the key issues surrounding climate change today, and
    what actions are being taken globally to address them?

RESPONSE:  ## Key Issues Surrounding Climate Change Today:
```

```
**1. Rising Global Temperatures:**

* **Consequences:** More frequent and intense heatwaves, droughts,
    wildfires, sea-level rise, melting glaciers and ice sheets, ocean
    acidification.
* **Action:** Transitioning to renewable energy sources, improving energy
     efficiency, reducing deforestation, carbon capture and storage
    technologies.

**2. Extreme Weather Events:**

* **Consequences:** Displaced populations, infrastructure damage,
    economic losses, food insecurity.
...
```

