# OpenReview forum: "LLM Neurosurgeon: Targeted Knowledge Removal in LLMs using Sparse Autoencoders"
_ICLR.cc/2025/Workshop/BuildingTrust — BuildingTrust_

### Official Review · Reviewer_RbPG · 2025-02-28
**Review of "LLM Neurosurgeon: Targeted Knowledge Removal in LLMs using Sparse Autoencoders"**

**Rating:** 7
**Confidence:** 5

**Review:**

## **Summary**

This paper proposes a novel method called Neurosurgeon, which uses sparse autoencoders (SAEs) to identify and suppress specific features activated when certain topics are present in the input prompt. The method involves generating synthetic data pairs one containing the target topic and one without using Gemini. These pairs are then fed into the model to calculate the activation frequency score, which helps in discovering the relevant features. By setting these features to zero, this method effectively prevents the model from generating outputs related to the targeted topics.


## **Strengths & Weaknesses**

### **Strengths**
- The paper is well-written and well-organized, presenting its methodology and results clearly.
- The authors employ two different methods for feature discovery.
- The proposed method is efficient as it does not require updating model parameters, reducing computational overhead.
- The evaluation covers both the target concept and related concepts, demonstrating that the targeted topic is successfully suppressed while maintaining the general performance of the model.

### **Weaknesses**
- While the paper evaluates both the target topic and related topics, a broader set of topics needs to be tested to strengthen the evaluation.
- It would be beneficial to test the method on removing harmful concepts and compare the results with other techniques mentioned in the abstract and introduction, such as DPO, RLHF, and unlearning methods.
- The paper lacks statistical analysis regarding the number of features that need to be clamped for different concepts, including both broad and specific topics. Such insights would help understand the applicability of proposed method across various domains.

---

### Official Review · Reviewer_iPCc · 2025-03-01
**This paper addresses computational concerns of RHLF and solutions such as prompt engineering that are hard to iterate. They introduce Neurosurgeon a procedure that uses (pre-trained) sparse autoencoders to identify and remove specific topics from a language model's internal representations.**

**Rating:** 5
**Confidence:** 3

**Review:**

Strengths:

This paper discusses addresses an important topic in AI safety. Their approach is creative and provides clear advantages (Flexibility, Precision and Computation Efficiency) against other traditional safety techniques such as RLHF.  They show consistent improvement in measured metrics (for eg. Coherence and Compliance) over the baseline.

It would be interesting if they compared to other methods apart from prompting the model to avoid a topic. Are there any RLHF or other adaptation techniques that this method can be compared to?

Weaknesses:

I have a few concerns about their method:
(1) how about how generalizable this approach can be? It would be interesting to see results on other models.
(2) Also, how expensive is the hyper-parameter tuning process? Do you need to figure out which layer etc is better for each feature?  Is it more expensive than what can be done with prompt engineering or RLHF?
(3) Can this method be used for implicit biases? For eg scenarios where religion can be mentioned but the LLM responses cannot contain nuanced and harmful portrayals of various religions.

---

### Decision · Program_Chairs · 2025-03-01

Accept